# Influence of Job Insecurity on Musculoskeletal Disorders: A Mediation Model with Nursing Aides

**DOI:** 10.3390/ijerph20021586

**Published:** 2023-01-15

**Authors:** Laura Mateos-González, Julio Rodríguez-Suárez, José Antonio Llosa, Esteban Agulló-Tomás, Juan Herrero

**Affiliations:** 1Department of Health Science, University of Oviedo, 33006 Oviedo, Spain; 2Department of Psychology, University of Oviedo, 33003 Oviedo, Spain; 3Department of Social Education, Padre Ossó Faculty, University of Oviedo, 33008 Oviedo, Spain

**Keywords:** musculoskeletal diseases, job insecurity, psychosocial risk factors, mediation analysis, occupational health

## Abstract

Work-related musculoskeletal disorders are some of the most prevalent diseases in the world. They have a multifactorial aetiology encompassing ergonomic and psychosocial risk factors. The aim of this study is to analyse the way job insecurity and physical workload are linked to musculoskeletal disorders, as well as the mediating role other psychosocial work risks can have on this potential relationship. A parallel mediation path regression analysis was designed using a sample of 457 nursing aides. The influence job insecurity and physical workload has on the onset of musculoskeletal symptoms together with the variables which mediate this relationship were examined. The results prove that both independent variables explain the onset of musculoskeletal symptoms in these professionals. The influence exerted by job insecurity is mediated by the social support received at work and the demands of the job. However, when analysing the physical workload, the social support received at work is not relevant as a mediator in this relationship. Job insecurity and physical workload are significant variables on the occurrence of musculoskeletal symptoms. The data obtained supports the need to focus on physical and psychosocial factors in order to prevent these disorders.

## 1. Introduction

Musculoskeletal disorders (MSDs) are some of the most prevalent diseases. They comprise a wide range of inflammatory and degenerative conditions affecting muscles, tendons, ligaments, joints, peripheral nerves, and support structures such as intervertebral discs [1].

According to the World Health Organization [2], approximately 1.7 billion people in the world suffer from MSDs, making it the biggest cause of disability globally. When MSDs are caused or aggravated by work, they are called work-related musculoskeletal Disorders (WRMSDs). The countries with higher earnings are the most affected by these disorders, which are the main cause of sick leave in developed countries [3]. Given their high prevalence, WRMSDs are a public health issue with high economic costs, both direct (medical treatments, sick leave, or compensations) and indirect (reduction on work productivity).

Despite being such a prevalent clinical entity, there is a lack of standardised diagnostic criteria, partly due to the wide variety of symptoms and signs reported by affected individuals. Identifiable and well-classified pathologies such as carpal tunnel syndrome or spinal disc herniation constitute only a small part of all reported MSDs.

The reliability of clinical assessment is variable, and often self-reported symptoms or functional disability are a better measure of the impact of the disease on patients. However, although symptoms are diverse and not always related to specific clinical findings, in the absence of “gold-standard” assessment techniques, musculoskeletal symptoms (MSSs) are used as outcome measures in the field of MSDs research as long as they are obtained in a standardised and reproducible manner [1].

MSDs have a multifactorial and, therefore, complex aetiology. There are individual factors, such as age, sex, body mass index, socioeconomic and educational level, health habits, or years on the job.

There are also occupational factors. The influence of physical and ergonomic risks is proven: having and holding an awkward posture at work, performing repetitive movements and handling loads (heavy loads and/or loads held over time) require excessive muscular activity which has an effect on the musculoskeletal system [4,5].

On the other hand, for many years now, occupational health has addressed these diseases not only from a physical point of view, but also from a psychosocial angle, considering the psychosocial risk factors as causes/aggravating factors of MSDs [6]. 

These psychosocial risk factors are defined as “Those aspects of work design and the organisation and management of work, and their social and environmental contexts, which have the potential for causing psychological, social or physical harm” [7] (p. 3).

There are several theoretical models which explain the impact psychosocial risk factors have on physical health. The “job strain model” [8] does this from a work stress viewpoint which happens as a result of an imbalance between the psychological demands of the job and the level of control an employee has over them. In this way, high demand combined with low control produces a stressful situation which, in turn, can have a negative impact on health. This theoretical model also includes social support from work colleagues and supervisors as a moderating element minimising the risk.

Due to the current work context, characterised by the insecurity and instability of jobs, emerging psychosocial risks have arisen, such as job insecurity (JI). JI is defined as the individual subjective anticipation of an involuntary loss of work [9]. Because of the increasing pressure on the efficacy and competitiveness of workers, decades ago Ganster et al. [10] identified JI as one of the main psychological demands faced by employees.

The perception of JI is a subjective reflection, but it is based on the employee’s objective situation. Therefore, it is associated with factors such as the unemployment levels and social policies of a country, the type of sector (public or private), or the type of contract [11].

JI implies a state of uncontrollability, due to its involuntary nature, and uncertainty about the future. While job loss is an immediate experience, JI entails a prolonged situation [9]. This makes it act as a chronic stressor that, additionally, does not allow the subjects experiencing it to make decisions or act (for example, by actively looking for work) [11]. Therefore, scientific research suggests JI poses a threat to health similar to unemployment itself [12], having a negative impact on physical [11] and mental health [13].

Depending on the presence of risk factors associated with MSDs, there are specific occupational groups more prone to suffer from these diseases. Thus, healthcare workers are engaged in work characterised by high physical and psychological demands. Additionally, due to the current contract flexibility and job insecurity affecting the sector, they suffer from high levels of job uncertainty [14]. As a result of the interaction of these and other risk factors, there is a high prevalence of MSDs in this sector [15]. 

There are complex interactions in the effects physical and psychosocial stressors have on musculoskeletal problems. The general aim of this study is to analyse various risk factors associated with the presence of MSSs, as well as the underlying mechanisms causing them, using as a sample an occupational group from the healthcare sector highly affected by these disorders: nursing aides.

In order to perform this, the relationship between the job insecurity and physical workload variables and suffering MSSs was analysed. Then, the mediating role of other psychosocial risks on this possible relationship was considered.

Although the association of psychosocial risk factors with MSDs has been widely studied in the scientific literature, this study addresses JI more precisely by collecting information with a specific scale and providing a mediation analysis that clarifies the correlation mechanisms between the variables. 

Two hypotheses are devised: Job insecurity and physical workload are relevant variables for the presence and development of MSSs; Psychosocial risks (measured in terms of job demands, social support at work, and job control) have a mediating role in the relationship described in Hypothesis 1. The parallel mediation path model hypothesised is shown in Figure 1.

## 2. Materials and Methods

### 2.1. Sample

The sample taking part in the study was comprised of 457 active nursing aides from residential care homes for the elderly in the Principality of Asturias (Spain). 

The Asturian Autonomous Organization of Residential Establishments for the Elderly (Organismo Autónomo de Establecimientos Residenciales para Ancianos de Asturias [ERA]) is an administration attached to the Department of Social Rights and Welfare (Consejería de Derechos Sociales y Bienestar) of the Asturian Autonomous Government (Spain) consisting of 35 centres spread across the region. This study was carried out across all centres, which currently have a combined total of 3378 residents.

The sampling method for the study was non-probability convenience sampling. Of the organisation’s 1172 active workers, 457 completed the questionnaire (38.9% response). The sample was composed of 431 women and 26 men. The total average age of the sample was 49.69 years (9.73) with an age range between 22 and 66 years. A total of 49.7% of the women had a permanent contract of employment and 40.3% a temporary contract. Among the men, 38.5% had a permanent contract and 61.5% a temporary one. These data are shown on Table 1. Further demographic data on the sample are available in Appendix A.

### 2.2. Instruments

The battery of tests used to gather the information necessary for this research consists of the following tests:

#### 2.2.1. Nordic Musculoskeletal Questionnaire (NMQ)

Musculoskeletal symptoms were measured using this standardised and self-administered questionnaire [16], which is widely used in occupational health. It identifies musculoskeletal symptoms in nine parts of the body: neck, shoulders, elbows, writs/hands, upper back, lower back, hips/thighs, knees, and ankles/feet. For each of these areas, a dichotomous answer (yes/no) is to determine the presence of symptoms over the previous 12 months, the previous 7 days, and the disability caused by those symptoms.

The coding method proposed by Pinheiro et al. [17] is used to obtain a total score regarding the level of musculoskeletal affect from the data gathered by the questionnaire. It uses a 0-4 Likert scale for each part of the body. A score of 0 indicates a total absence of symptoms; a score of 1 means they only suffered from symptoms within the previous 12 months; a score of 2 indicates both symptoms within the previous 12 months and in the past 7 days without any disability; a score of 3 means symptoms within the previous 12 months plus a disability caused by them; finally, a score of 4 indicates symptoms during the previous 12 months and the past 7 days, as well as a disability caused by these symptoms. The questionnaire’s total score ranges from 0 to 36.

#### 2.2.2. Job Insecurity Scale-8 (JIS-8)

The JI variable was evaluated using this 8-item questionnaire which measures cognitive (plausibility of job loss) and affective (fear of job loss) dimensions proposed by Borg and Elizur [18]. It is a 1–5 Likert-style scale; the first 4 items correspond to the cognitive dimension and the last 4 to the affective dimension. The higher the score, the higher the perception of job insecurity.

This paper used the Spanish validation of the scale [19] that offers a total score with a reliability of α = 0.88. 

#### 2.2.3. Job Content Questionnaire (JCQ)

It is a questionnaire that measures different psychosocial risk factors using Karasek and Theorell’s demand-control-support model [8]. This study used the Spanish validation [20], which is comprised of 22 items grouped in 3 dimensions: psychological demands at work (7 items), which gather information about the amount of work, intellectual demands and time pressure; control over the work (7 items), which concentrates on decision making possibilities, creativity, and development of one’s own abilities; and social support at work (9 items), which includes support received from peers and/or bosses. They are 1–4 Likert scales with an internal consistency of α > 0.7. 

#### 2.2.4. Physical Workload Questionnaire (PWQ)

This self-reporting questionnaire, developed by Hollman et al. [21] using a population of workers from geriatric centres, provides an objective estimation of the physical workload from a subjective assessment on the mean frequency of adopting specific postures and manual handling of loads during a regular workday.

It consists of 19 items grouped in different blocks: 5 items describe different trunk postures, 3 items describe arm postures, 5 for the legs, 6 items describe load handling in three ranges (<10 kg, 10–20 kg, >10 kg) and 2 posture modalities: weight with the trunk upright and weight with the trunk at a 60° incline.

There are pictograms representing the items and a 5-point Likert-style scale (0 = never to 4 = very often) is used to answer.

From this information and using a biomechanical theoretical model [22] which describes the compressive forces on the lumbar spine during work activities, it builds a physical workload index (Physical Workload Questionnaire Index) which is calculated through a weighted sum of the 15 items describing examples of physical overload.

#### 2.2.5. Sociodemographic Data

The sociodemographic section of the questionnaire gathers general information about the interviewee: age, gender, and nationality, as well as information about the workplace: geographical location and type of contract.

The battery also included data about quality of life in relation to health through the EuroQuol 5DL (EQ5DL) questionnaire [23].

### 2.3. Procedure

The participants were provided the tests from the battery on the following order: NMQ; EQ5DL; JQC; JIS-8; PWQ, and finally, a sociodemographic and socio-occupational block. It is a self-administered battery filled using pencil and paper. The answers were encoded by the research team into a data matrix generated with SPSS v25 (IBM Corp., Armonk, NY, USA and analysed with JASP 0.16.4. The database files were stored in servers owned by the University of Oviedo. Every participant in the study took part on a voluntary basis, having previously been informed both of the objectives of the research analysis and of the guarantees of anonymity provided by this research. In order to take part in the study, the participants provided explicit informed consent.

This study was presented to and received the approval of the Research Ethics Committee of the Principality of Asturias, Spain (Comité de Ética de la Investigación del Principado de Asturias), (code CEImPA 2021.360). It also has explicit permission from the board of the Asturian Autonomous Organization of Residential Establishments for the Elderly (Organismo Autónomo de Establecimientos Residenciales para Ancianos de Asturias [ERA]), an administration attached to the Department of Social Rights and Welfare (Consejería de Derechos Sociales y Bienestar) of the Asturian Autonomous Government (Spain). Additionally, the study was developed in accordance with the principles of the Declaration of Helsinki.

### 2.4. Data Analysis

To test the hypothesis stated, a mediation analysis was carried out with the sample described using a cross-sectional design. The first part of the analysis involved a descriptive analysis and Pearson correlation analysis (CI 95%) between the variables included in the hypothetical model stated. None of the Pearson correlations exceeded 0.70, with multiple co-linearity between the variables analysed being unlikely [24]. Then, the parallel mediation analysis was carried out through path analysis regressions using the following variables: independent variables (IVs): job insecurity and physical workload; dependent variable (DV): musculoskeletal symptoms; and parallel mediations with the variables included in the JCQ’s three measures of psychosocial risks: social support (M1), job demands (M2), and job control (M3). The regression parallel mediation analysis was carried out with 5000 bootstrap samples, taking into account a confidence interval (CI) of 95%. The β calculated through the bootstrap sample method matched the direct estimate of the model, which proves that the results are robust to possible sample heterogeneity effects [25]. The total effects of job insecurity and physical workload on the variable MSSs were extracted. The total indirect effect of each of the mediation paths generated was also extracted: on the first one, the IV was job insecurity together with the three mediating variables (the three subsets from the JCQ), and on the second the IV was the physical workload. The direct effect of each independent variable on the MSSd was also calculated, without taking into account the weight of the moderating variables. Finally, an analysis of the indirect effects of each of the mediation paths generated in the analysis was carried out. The estimators of indirect effects were compared to observe which variables showed the highest relevance to the mediation process stated. The results from the analysis allowed to hypothesise dependency relationships, but being a cross-sectional design, it did not show the possibility of determining these dependency relationships. Additionally, the sample used corresponds to a healthcare professional profile, which cannot be directly generalised to any other professional activity.

The parallel mediation process followed the design formulated by the research carried out by Zeytinoglu et al. [26], and Ordóñez-Camblor et al. [27], and Hayes’ [28] methodology for the analysis of mediation and moderation. All analyses were developed using Jasp Ver. 0.16.4. (JASP Team, Amsterdam, The Netherlands).

## 3. Results

The results section of the study has three subsections. The first part is comprised of the descriptive results from the variables and their correlations. The remaining subsections include the data needed to study both hypotheses. 

### 3.1. Descriptive Analysis and Correlations

Table 2 shows the mean and typical deviation for the variables of musculoskeletal symptoms (measured by the NQM); physical workload (measured by the PWQ); job insecurity (measured by the test JIS-8); and psychosocial risks, measured by the three dimensions of the JCQ test: job demands, job control, and social support at work. The results are disaggregated by gender (male and female).

The highest correlation between social support at work and job control was r = 0.258 (Table 3). As stated by way of an assumption in the analysis section, these correlational results are indicative of the absence of multi-collinearity between the variables. All the variables correlate significantly (CI = 95% or CI = 99%) with the musculoskeletal symptoms measure, with the exception of the psychosocial risk of job control (r = 0.045). Finally, all social support correlations are inverse, except for job control. As for the social support measure it is assumed that the higher the score, the higher the support.

### 3.2. Hypothesis 1

The analysis of residual covariances allows the assumption of homoscedasticity to be accepted (β = −438, *p* > 0.05), thus continuing with the parallel mediation model [29]. 

Regarding Hypothesis 1, the results obtained from the mediation model show that the total effect is significant for both independent variables. For job insecurity (IV), high scores evince an increase in scores on MSSd (DV) (β = 0.179, *p* < 0.01). The same relationship is established on total effects when the IV relates to physical workload (β = 0.171, *p* < 0.01) (Table 4). When analysing the direct effects of both independent variables on MSSs, they are significant, which indicates there are more unknown mediating variables for this relationship. The direct effect for the job insecurity IV is β = 0.124 (*p* < 0.05) and for physical workload IV 0.123 (*p* < 0.01) (Table 5). Overall, the mediation model presented, with the two predicting and mediating variables, explains the 12.7% (R2) variance on the MSSs scores. The analysis of the total effect estimators shows that the relevance of both variables on musculoskeletal symptoms is similar in the model.

### 3.3. Hypothesis 2

Although the direct effects of the independent variables on the MSSs score remain significant in the mediation model, some of the indirect effects calculated in the parallel mediation are also significant. Therefore, a mediating effect can be seen on the relationship established, which allows to gather the results needed to analyse the second hypothesis. 

In the case of path analysis regressions where job insecurity is the IV and the MSSs score is the DV, the total indirect effects yield significant value (β = 0.054, *p* < 0.05). When analysing the indirect effect of each parallel mediator of the psychosocial risks, it is observed that the estimators of social support at work (M1) and job demands (M2) are significant. The indirect effect of job control (M3) is not significant (β = 0.0004, *p* > 0.05). Figure 2 shows that social support is a protective factor against MSSs for people with a high score on job insecurity (β = 0.027, *p* < 0.05). However, a high score on job demands means a risk factor regarding MSSs when job insecurity is scored high (β = 0.027, *p* ≤ 0.05). Both mediating variables account for 50% of the weight of the total indirect effects between job insecurity (IV) and the MSSs score (DV). Therefore, their relevance for understanding this relationship is similar. The mediating effect of both variables represents 30.16% of the total effect of job insecurity (IV) on the MSSs score (DV), which helps understand why the direct effects of job insecurity are significant in the mediation model. 

When the parallel mediation is analysed with the physical workload estimator, the indirect effects are also significant (β = 0.049, *p* < 0.01). However, in this case the relevant mediator is only the job demands (β = 0.041, *p* < 0.01). High demands increase the possibility of employees on jobs with a physical workload suffering from musculoskeletal symptoms. In this case, the mediation path with social support at work as a mediating variable is not significant (β = 0.008, *p* < 0.01). The total indirect effects account for 28.66% of the total effect in the relationship between physical workload (IV) and the MSSs score (VD).

## 4. Discussion

The aim of this study was to clarify the relationship between the occurrence of MSSs and different factors of occupational risks. 

Based on the previous literature [4,5], this study obtained similar results regarding the link between physical load and MSDs. It also provides relevant information on job insecurity (JI), a psychosocial risk less researched specifically in relation to musculoskeletal health. Even though the effects of JI on physical and mental health have been widely researched in scientific literature [11,19], the specific influence of this risk factor on MSDs has received little empiric attention until now [26,30]. This study aims to fill this gap by providing statistically substantiated data which allows to continue developing this line of research.

There are various theoretical models which explain the link between the presence of JI and its negative consequences for the health and wellbeing of employees. Jahoda’s latent deprivation model [31] and Conservation of Resources theory [32] define the role of work not only as an income generator but as a way for people to obtain and maintain basic resources such as social relationships outside of the family circle, the opportunity of developing or acquiring social status. JI threatens the fulfilment of these needs.

Self-Determination Theory [33] defines three basic psychological needs: autonomy (freedom of choice), belonging (having significant social relationships), and competence (influencing their environment and achieving the desired outcomes). Experiencing JI hinders the achievement of these three needs due to its uncontrollable and unpredictable nature.

Along the same lines, Warr’s Vitamin Model [34] suggests nine environmental vitamins for maintaining good psychological health, among which are the opportunity for interpersonal contact or a valuable social status. Many of these vitamins are compromised or threatened by the perception of JI.

On the other hand, the psychological contract theory [35] explains that workers who perceive the JI experience as a breach of the psychological commitment inherent in the contract with the company feel that their loyalty and dedication is not being rewarded and this impacts negatively their affective organizational commitment.

All these theoretical models lead to the employees experiencing stress, burnout, and low job satisfaction, with their consequent implications for psychological and physical health.

The work carried out on this paper follows the research performed by Zeytinoglu et al. [26,30], both from the point of view of similarity between the sampling (home-care workers and nursing aides from geriatric centres) and the mediation analysis of the possible risk factors involved in the occurrence of MSDs.

Hypothesis 1 is consistent with previous studies which links the presence of MSDs to a higher physical load at work [4,5]. The increase in physical load at work means a muscular and biomechanical overload which leads to the deterioration of musculoskeletal health.

The direct link between the presence of JI and MSDs is also similar to previous studies [26,30,36]. Although the mechanism linking job insecurity to musculoskeletal symptoms is not known, a hypothesis was formulated which suggests that workers with unstable jobs are more likely to carry on working after developing musculoskeletal symptoms, for fear of being fired: by refusing to reduce their exposure by changing jobs, these workers can suffer from symptoms increasingly more severe symptoms [36]. It also was suggested that work-related stress can happen both as a result of a change in the nature of the work and of insecurity in the labour market and that this type of stress can easily cause MSDs [26].

This leads to Hypothesis 2. While Zeytinoglu et al. [26] used work-related stress as the mediating variable, the present study included other psychosocial risk factors which, in turn, have also been proposed by scientific literature as catalysts for work-related stress. Thus, Karasek and Theorell’s demand-control-support model [8] describes work-related stress as the result of combining these three factors.

The increase on job demands, which mean bigger workload and time pressure, acts as a mediator on the link between both workload and JI and the presence of MSDs. This is consistent with previous studies which proved the influence this psychosocial risk factor has on MSDs [37] and which in this case increases the negative effects of both independent variables.

As a decisive psychosocial resource for improving psychological health [34] and work-related stress [8], social support acts as a mediator easing the negative effects of JI on musculoskeletal health. This is compatible with previous studies in which social support acts as protection against MSDs [37]. However, in this case, it does not act as a mediator on the influence of physical workload. 

In contrast to previous research [38], this study did not prove that the job control variable has a mediating role on the negative effects of JI. This can be due to the individual characteristics of the job function analysed. By having fully outlined tasks and not much decision-making capacity, there is no variability of this risk factor between subjects.

### Implications for Practice

In the current geopolitical framework, which presumably has an impact as a global economic recession, it is imperative that businesses are aware of the negative impact this economic downturn has on employees’ health. Situations such as downsizing, mergers, or restructuring within a company can lead to an increase in physical workload, psychological demands, and JI.

Within the field of occupational health, individual and company-wide interventions should be carried out with the aim of managing physical and psychosocial work stressors in order to improve workers’ health and wellbeing, thus preventing the onset of MSDs.

Individual interventions include promoting postural hygiene at work to mitigate the impact of physical workload [39] or developing employees’ skills, such as resilience, to lessen the negative impact of JI [40].

At the company level, action should be taken on biomechanical factors to reduce the impact of the physical load through the ergonomic adaptation of both furniture and tasks. Interventions for reducing JI can include measures such as increasing the workers employability through company-funded training in order to improve their work skills or manage clear and timely communications with the employees in matters of company restructuring or downsizing.

Other suggestions include promoting interpersonal relationships within the company, both among employees and between workers and managers to improve social support which acts as a protector of health, and participative decision making which lessens the negative effects of the psychological demands caused by JI [38] situations.

These suggestions should be considered without forgetting macrolevel interventions such as improving social protection policies or the development of more active labour policies: “adapting the human capital of the unemployed to labour market demands by providing vocational training and stimulating job search efforts also decrease perceptions of job insecurity, as well as creating and/or subsidizing new jobs” [11] (p. 122).

## 5. Conclusions

This study analyses the relationship between MSSs and their risk factors, reinforcing the available evidence on the link between physical workload and these pathologies and providing data on job insecurity, a less researched risk factor in relation to MSDs. In this way, new research lines are promoted in the field of occupational health. The mediation analysis provides relevant information on the factors that may intervene in the process.

The results from this study show that both physical workload and JI are linked to the presence of MSSs. Additionally, high job demands increase the possibility of suffering from musculoskeletal symptoms in employees in jobs with high physical workload and who experience JI. Social support at work acts as a protector against MSSs in people with a high JI score, but not for those with a high physical workload.

As a limitation, it should be noted there is a need to develop future research with a longitudinal design. Additionally, the sample used is comprised exclusively of healthcare workers, so it is not generalisable to other occupational groups.

Another limitation of this study is the use of self-administered questionnaires that use subjective measures. Although this type of tool is used in occupational health research due to its feasibility and low cost, future studies using other more objective measures, such as clinical evaluation or diagnostic tests are necessary.

Faced with pathologies as prevalent and disabling as MSDs, it is essential to focus, as it was performed in this study, on the analysis of the mechanisms causing them, with a view to provide statistical data which support the development of the necessary preventive interventions and action policies.

## Figures and Tables

**Figure 1 ijerph-20-01586-f001:**
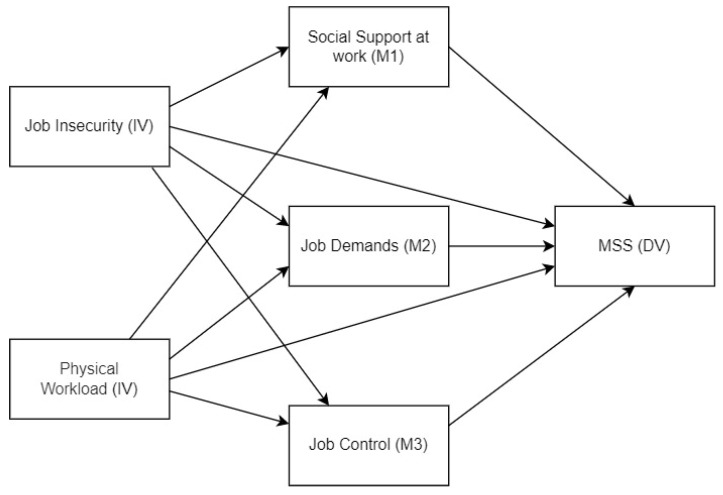
Hypothesized parallel mediation model. IV: Independent Variable; M: Mediator; MSSs: Musculoskeletal Symptoms.

**Figure 2 ijerph-20-01586-f002:**
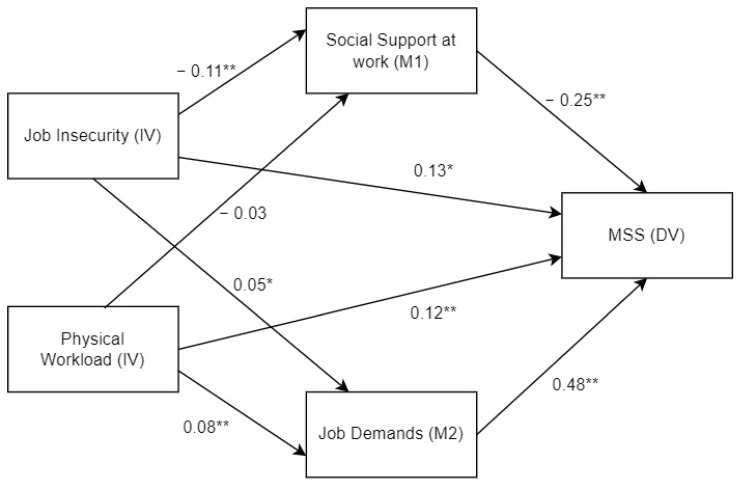
Resulting parallel mediation model. * *p* < 0.05, ** *p* < 0.01.

**Table 1 ijerph-20-01586-t001:** Sample characteristics.

		SEX (Mean Age)		
Type of Contract		Female	Male	Total
Permanent	Count	214	10	224
	% within column	49.7 %	38.5 %	49 %
Temporary	Count	217	16	233
	% within column	50.3 %	61.5 %	51 %
Total	Count	431 (49.90)	26 (49.35)	457 (49.69)
	% within column	100 %	100 %	100 %

**Table 2 ijerph-20-01586-t002:** Descriptive analysis.

		Mean	Std. Deviation	Minimum	Maximum
Total_NMQ	Female	13.66	7.82	0.00	36.00
	Male	10.35	9.53	0.00	30.00
	Total	13.74	7.95	0.00	36.00
PWQ_Index	Female	39.56	10.36	10.15	56.17
	Male	37.80	12.10	18.17	56.17
	Total	39.49	39.49	10.15	56.17
JIS_TOTAL	Female	19.18	5.73	8.00	38.00
	Male	20.04	7.34	8.00	36.00
	Total	19.24	5.84	8.00	38.00
JCQ_Demand	Female	18.00	3.24	6.00	24.00
	Male	17.15	3.84	11.00	24.00
	Total	17.95	3.28	6.00	24.00
JCQ_Control	Female	18.60	3.42	7.00	28.00
	Male	18.15	3.22	10.00	24.00
	Total	18.57	3.40	7.00	28.00
JCQ_Support	Female	25.48	4.47	9.00	36.00
	Male	25.00	5.25	13.00	34.00
	Total	25.47	4.52	9.00	36.00

**Table 3 ijerph-20-01586-t003:** Pearson correlation between variables.

Variable	Total_NMQ	PWQ_I	JIS_TOTAL	JCQ_Demand	JCQ_Control	JCQ_Support
1. Total_NMQ	—					
2. PWQ_Index	0.214 ***	—				
3. JIS_TOTAL	0.136 **	0.028	—			
4. JCQ_Demand	0.251 ***	0.284 ***	0.108 *	—		
5. JCQ_Control	0.045	0.070	0.009	0.250 ***	—	
6. JCQ_Support	−0.182 ***	−0.069	−0.142 **	−0.074	0.258 ***	—

Note. * *p* < 0.05, ** *p* < 0.01, *** *p* < 0.001.

**Table 4 ijerph-20-01586-t004:** Total effects path analysis regressions.

		Estimate	Std. Error	z-Value	*p*
JIS_TOTAL → Total_NMQ		0.179 **	0.063	2.861	0.004
PWQ_Index → Total_NMQ		0.171 **	0.035	4.931	<0.001

Note.** *p* < 0.01.

**Table 5 ijerph-20-01586-t005:** Direct effects, total indirect effects, and indirect effects of parallel mediation with path analysis regressions.

	Estimate	Std. Error	z-Value	*p*
IV: Job insecurity; DV: MSSs				
Direct Effect JIS_TOTAL→ Total_NMQ	0.125 *	0.062	2.032	0.042
Total indirect effect	0.054 **	0.019	2.854	0.004
JIS_TOTAL → JCQ_Support→ Total_NMQ	0.027 *	0.013	2.148	0.032
JIS_TOTAL → JCQ_Demand→ Total_NMQ	0.027 *	0.014	1.922	0.055
IV: Physical workload; DV: MSSs				
Direct Effect PWQ_Index→ Total_NMQ	0.123 **	0.035	3.497	<0.001
Total indirect effect	0.049 **	0.013	3.707	<0.001
PWQ_Index → JCQ_Support→ Total_NMQ	0.008	0.006	1.364	0.173
PWQ_Index→ JCQ_Demand→ Total_NMQ	0.041 **	0.012	3.464	<0.001

Note. * *p* < 0.05, ** *p* < 0.01.

## Data Availability

The data presented in this study are contained within the article.

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
