# Peer review of "Influence of Job Insecurity on Musculoskeletal Disorders: A Mediation Model with Nursing Aides"

_ijerph, 2023, doi:10.3390/ijerph20021586_

Round 1
Reviewer 1 Report
The theme of the article is very actual. The article, overall, shows an interesting study, but must be revised according to the following comments.
1. Formatting and language issues:
The article overall has a number of grammatical issues and phrasal constructions that are not adequate. It would likely benefit from a round of editing prior to a re-submission.
For example:
line 65, change “competitive” to “competitiveness”
In figure 1, the acronym for Musculoskeletal disorders is not correct. Presents MDS instead of MSD. Please, correct the acronym. Take into consideration the recommendations below regarding the correct use of the acronym for musculoskeletal symptoms.
Line 237, correct “evince”
Line 256 – correct the sentence.
1. Title and Abstract:
The Title is clear and the abstract includes the relevant information that frames the reader.
2. Introduction:
The introduction is well structured and presents the relevance of the theme. Presents an interesting overview of the state of art and the main theories that support the aim of the study and the hypothesis.
Consider revising the phrase “As a result, there is a high prevalence of MSDs in this sector [15].” in line 81, because the high prevalence of MSD is the result of several factors and their interaction.
Ends with the presentation of the aim and the hypothesis of the study, clear and adequate.
3. Materials and methods:
Regarding the instruments, the Nordic Questionnaire is not used to measure MSD, but to measure musculoskeletal symptoms (MSS) that may be related to MSD. This paragraph should be revised.
The presentation of the tool has implications for the definition of the aims of the study because the outcome is not MSD but MSS. And it also implies an explanation in the introduction about the relationship between MSD and MSS, in order to justify the title of the article. It will be necessary to review the abstract and the keywords.
The statistical analysis is adequate and well-described.
All over chapter 3 the acronym MSD should be changed to MSS.
Regarding the description of the independent variable in line 201, the designation of ergonomic risks should be avoided because you are measuring physical workload. The definition of ergonomics in itself is in opposition to the use of the expression ergonomic risks.
4. Results:
In line 220, following what was recommended above, please consider changing “the variables musculoskeletal disorders” to musculoskeletal symptoms and use it along with the presentation of the results in the article.
Globally, the results are properly presented.
5. Discussion:
The first part of the discussion lists several theories, without a deeper view. It could be more interesting to choose the most adequate and to make a more profound analysis.
Nevertheless, the discussion is adequate and relevant.
6. Conclusion:
The conclusion reflects the contribution of the work to occupational safety and health knowledge and application.
Reviewer 2 Report
The article deals with the very important problem of identifying risk factors for the occurrence of musculoskeletal disorders in employees. I noticed some ambiguities in it that require correction or a broader discussion:
1. In the abstract, the authors write that job insecurity and physical workload are significant variables on the onset and development of musculoskeletal disorders. This statement is unjustified, because the study did not determine the onset or development of disorders, only their occurrence.
2. The tools that were used in the study were a questionnaire for assessing musculoskeletal symptoms (Nordic Musculoskeletal Questionnaire - NMQ), job insecurity (Job Insecurity Scale-8), different psychosocial risk factors using Karasek and Theorell's demand-control-support model (Job Content Questionnaire) and the physical workload (Physical Workload Questionnaire). All these questionnaires use subjective ratings. The use of only subjective assessments of a phenomenon is not a mistake, but the question always arises to what extent such assessments reflect reality. This is especially true of the Job Insecurity Scale - it was not checked whether the surveyed person, when answering the questions in this questionnaire, describes the objective situation in the workplace, or their own fears resulting from the difficulty of continuing work that is too hard for them and causes discomfort. The former may worsen and the latter may improve the correlation between Job Insecurity Scale score and musculoskeletal disorders.
3. The aim of the study was to verify the hypothesis "Job insecurity and physical workload are significant variables for the occurrence and development of MSDs." It is known that there are many causes of MSDs, and not all of them are related to physical exertion at work, and the importance of occupational and non-occupational risk factors may vary depending on the location of MDS. However, the role of non-occupational risk factors for MSDs has been completely overlooked.
Reviewer 3 Report
Estimated Authors of the paper "Influence of job insecurity on musculoskeletal disorders: a mediation model with nursing aides",
I've read with great interest your paper. MSD represent a main issue for professionals involved in the management and prevention of work related disorders, particularly amon healthcare professionals. Risk factors for MSD are not only represented by physical ones (i.e. the actual patient handling) but encompass all the work environment, including psychological factors.
In this study, Laura Mateos-González et al identify Job insecurity and physical workload as significant effectors on the onset and development of musculoskeletal disorders. As explained across the text, "the influence exerted by job insecurity is mediated by the social support received at work and the demands of the job": all of the aforementioned results are both not unexpected according to the previous reports (it is clearly stated in the discussion) and interesting.
Therefore, from my point of view, this paper would deserve a full publication on IJERPH but some improvements are needed, and more precisely:
1) as this kind of study is not radically new, please highlight in the introduction and in the discussion why this study could improve our understanding of this specific topic, and which novelty was implemented by the Authors (either in the design or data analysis) that would add some information to our global understanding of MSD in healthcare settings;
2) Authors were able to retrieve several demographic information, but it is not shown across the main text. Please: a) add in Table 1 some information on age groups and seniority (the latter if available) of professionals, as this specific variables are deeply rooted in the development of MSD; b) provide as an annex table or supplementary table the detailed demographic data of participants;
3) please provide some information about the parent healthcare service: we're briefed that the recruited HCW were from "residential care homes for the elderly in the Principality of Asturias (Spain)": how many residential homes were inquired ? how many inmates have the sampled residential homes?
4) please provide some information about the potential number of targeted HCW, in order to calculate a dropout rate, that would guarantee some significant information about the actual representitivity of the sample.
